# Factors Predicting Outcomes of Supine Percutaneous Nephrolithotomy: Large Single-Centre Experience

**DOI:** 10.3390/jpm12121956

**Published:** 2022-11-25

**Authors:** Yasmin Abu-Ghanem, Luke Forster, Pramit Khetrapal, Gidon Ellis, Paras Singh, Rohit Srinivasan, Rajesh Kucheria, Anuj Goyal, Darrell Allen, Antony Goode, Dominic Yu, Leye Ajayi

**Affiliations:** 1Department of Urology, Royal Free Hospital, London NW3 2PS, UK; 2Department of Radiology, Royal Free Hospital, London NW3 2PS, UK

**Keywords:** supine, percutaneous nephrolithotomy, nephrolithiasis, outcome, predicting factors

## Abstract

Objective: Percutaneous nephrolithotomy (PCNL) is the treatment of choice for large renal calculi. The prone position has been considered the preferred position to obtain renal access. However, the supine position has recently gained popularity, which confers several potential advantages. The current study analyses the prognostic factors for successful supine PCNL procedures in a larger tertiary centre. Subjects: Prospective data were collected from all patients undergoing PCNL in the Galdako modified Valdivia position at our institution between February-2007 and September-2020. Surgical outcomes variables collected included: the rate of Endoscopic-combined intra-renal surgery (ECIRS), operative times, surgical effectiveness (no residuals <2 mm stone fragments) and complications. Results: A total of 592 patients underwent PCNL with a median age of 56 years (IQR: 42–67). The median stone size was 17 mm (IQR: 13–23). Of those, 79% of patients had an effective procedure. Stone size (*p* < 0.001), location (*p* < 0.001) and Guys-Stone Score (GSS) (*p* < 0.001) were associated with effectiveness. A Percutaneous nephrostomy tube was sited at the completion of the procedure in 97.3% of patients and a simultaneous double-J stent in 45.3%. Stent insertion was associated with larger stones (*p* < 0.001), the performance of ECIRS (*p* < 0.001) and higher GSS (*p* < 0.001). The overall complication rate was 21.7%. The main type of complication was an infection in 26.2 of the cases followed by the need for repeated nephrostogram in 12.7%. Conclusions: We demonstrate that PCNL in a high-volume centre is safe and efficacious in the Galdalko modified Valdivia position. Patients with smaller stones in the renal pelvis and a low GSS have the highest chance of a successful procedure.

## 1. Introduction

Currently, PCNL is the treatment of choice for large renal stones, staghorn calculi, or in stones occurring in kidneys with abnormal anatomy [1,2]. Traditionally, PCNL has been performed in the prone position, and the morbidity of the procedure has been well documented. However, the supine approach’s safety and efficacy and potential advantages likely explain the increasing use of the supine position for stone management over the last ten years [3]. Since the Valdivia position for supine PCNL was first described, it has been modified and refined by several authors to optimize both the surgical and anesthesiologic requirements. One of the most commonly used approaches is the Galdakao-modified Valdivia position [4]. Since introduced in 2007 [5], several authors have reported their experience with this position and confirmed its safety and high success rate [6,7,8].

Nevertheless, despite previously reported advantages, it has been pointed out that the supine position has its flaws, such as prolonged operative time in the case of larger stones. At present, current guidelines do not support one approach over the other. Therefore, controversy still exists regarding which position is safer and more effective for PCNL. In the current study, we aim to investigate the factors that may predict the success and safety of the supine position in a large single-centre cohort in an attempt to characterize better the patients who may benefit from this approach.

## 2. Materials and Methods

Data were collected on 592 consecutive patients undergoing PCNL at our institution between February 2007 and September 2020. A retrospective analysis was performed. All patients underwent preoperative assessment in a dedicated stone clinic including: clinical examination, routine laboratory tests, and CT imaging. All cases were discussed at a stone multidisciplinary team (MDT) meeting, that includes a consultant urologist and a radiologist. Demographic variables included: patients’ age, gender body mass index (BMI), Charlson comorbidity index (CCI) and history of renal surgery. Clinical and surgical variables included recorded: date of surgery, stone size (recorded as the maximal diameter for single stones and maximal diameter of the largest stone for multiple stones), indication (stone/encrusted stent), location (upper pole/middle pole/lower pole/renal pelvis/partial staghorn/ staghorn), and stone density (HU). In addition, stones were classified according to the Guy’s Stone Score (GSS) [9].

Three consultant endourologists carried out all operations in the Galdako-modified Valdivia position. A 3-Liter air-filled irrigation bag was used to elevate the patient’s flank to attain the optimal position. Supine PCNL was performed as previously described by this team [8]. Access was guided with ultrasound, fluoroscopy, or a combination of both by a dedicated uro-radiologist. All procedures were performed via a standard 30F tract. Tract dilatation was performed using a Nephromax (Boston Scientific, Marlborough, MA, USA) or Ultraxx (Cook Medical, Bjaeverskov, Denmark) balloon dilator.

When indicated, endoscopic combined intrarenal surgery (ECIRS) was carried out by an assisting surgeon using the digital Flex XC (Karl Storz, Tuttlingen, Germany) flexible ureterorenoscope. A 16F urethral catheter was inserted on free drainage, a 10F locking nephrostomy tube on open drainage was sited at completion of the procedure, and in selected cases, a retrograde or antegrade, Double-J stents were sited.

Operative parameters were recorded, including duration of surgery and patient radiation exposure. Standard postoperative care was carried out in a dedicated urology ward. The urethral catheter was removed on day 1 and the nephrostomy on days 1–2 depending on clinical judgment.

Surgical outcomes variables collected operative times, surgical effectiveness and complications. All patients deemed clear at the time of surgery were evaluated using non-contrast CT imaging at Six weeks postoperatively to assess stone clearance, as per standard local practice.

An effective procedure was defined as either residual fragments <2 mm or stone-free patients. In the case of significant residual stone detected intraoperatively, subsequent procedures were booked without updated cross-sectional imaging.

Complications were graded using a modified Clavien-Dindo Classification (CDC) [10].

Statistical analysis was performed using Statistical Package for Social Sciences (SPSS, Version 22.0, Chicago, IL, USA). Unless otherwise specified, data is given as median/interquartile range (IQR). Student’s *t*-test was used for the analysis of continuous variables, and the Chi-square test was used for the analysis of categorical variables. The variables that were included in the univariate analysis were included in a multivariate analysis using binary logistic regression, to evaluate the association of the different groups with outcomes, controlling for clinical variables.

## 3. Results

### 3.1. Demographics

A total of 592 patients underwent PCNL, including 347 (58.6%) men and 245 (41.4%) women, with a median age was 56 (IQR: 42–67) years and a median body mass index (BMI) of 28 (IQR: 24.8–32.5). The median CCI grade was 2 (0–4). Procedures were left-sided in 321 (54.2%), right-sided in 262 (44.3%), two patients underwent bilateral PCNL, and three patients underwent PCNL in a transplanted kidney. The median stone size was 17 mm (IQR: 13–23) with a median density of 1008 HU (IQR: 697–1342). The median GSS was 2 (IQR: 1–3). Calculi were located in the renal pelvis in 177 (30%) patients; 82 (13.9%) had a staghorn stone and 100 (16.9%) presented with multiple stones. Fourteen patients underwent surgery due to a forgotten encrusted stent. All procedures were performed by three surgeons (Table 1).

### 3.2. Access

In all, PCNL cases were performed in the supine position. Renal access was achieved with a single puncture in 565 (95.4%) cases, while 27 (4.6%) patients required a second puncture, mainly due to multiple calices involvement or staghorn stones (Table 2). Ultrasound alone was used for access imaging in 79 (13.3) cases, fluoroscopy alone in 8 (1.4%) cases, and a combination in the remaining 488 (82.4%) cases. Access was achieved via the lower pole in 333 (56.3%) cases, whereas upper and middle pole/ interpolar calix access were performed in 12.5% and 31.2% of cases, respectively.

In most cases (94.9%) the tract was dilated using a nephrostomy balloon catheter (NephroMax™ or Ultraxx™), whereas the remaining 5% were dilated via a metal telescopic dilator. Ultrasonic and pneumatic lithotripters were the most commonly used energy source in 73% of cases. Failed access occurred in 12 (2%). Of those, 6 (50%) stones were converted to flexible ureteroscopy (URS) and laser vaporization of the stone at the same setting, whereas the remaining 6 underwent delayed procedure. All 12 were excluded from further outcome analysis.

Simultaneous ECIRS was carried out in 164 (27.7%) cases (Table 3), with a flexible ureterorenoscope in 147 (24.8%) and a rigid scope in 17 (2.9%) cases. ECRIS was more likely to be performed in patients with multiple stones or encrusted stents. Moreover, a combined procedure was more commonly performed in larger stones yet with lower GSS (*p* < 0.05).

### 3.3. Outcomes

The median operative duration was 60 min (IQR: 40–84). Median fluoroscopy time and radiation dose were 592 (IQR: 370–817) 374.5 s and 108 (IQR: 95–120) mGy, respectively. The mean hospital stay was three days (IQR: 2–5).

A total of 399 (68.8%) patients had no residual stone on CT, 58 (10%) had <2 mm stone fragments, whereas 123 (21.2%) had remaining stone fragments larger than 2 mm.

A total of 79% of patients were deemed to have an effective procedure by the mentioned criteria. A PCN tube was sited at the completion of the procedure in 97.3% of patients. A simultaneous double-J stent was sited in 45.3% of cases. None of the patients underwent a “completely tubeless procedure” [11]. Most of those who received a stent had a stone in the renal pelvis, followed by staghorn and lower pole stones (26% and 17.1%, respectively). Moreover, stent insertion was associated with larger stones (21.3 mm vs. 18 mm, *p* < 0.001), the performance of ECIRS (64.4 vs. 35.6%, *p* < 0.001) and higher GSS (*p* < 0.001), (Table 4). Further analysis of the 428 patients who underwent PCNL without a combined URS revealed similar risk factors for stent insertion regardless to ureteroscopic access (Table 4). Stone size (HR:1.03, CI:[1.002–1.6], *p* = 0.035), high GSS (HR:1.3, CI:[1.0–1.7], *p* = 0.05), and ECIRS (HR:3.6, CI:[2.3–5.7], *p* = 0.00) remained significant on multivariant analysis.

In regard to complications, the overall complication rate was 21.7%. The main type of complication was infection in 26.2% of cases. Only 13 (2.2%) patients required blood transfusion. Median decrease in hemoglobin levels was 1.3 g/dL (IQR: 0.7–2.1). Only 5 patients (0.86%) required postoperative blood transfusion.

Major complications accrued in 6 patients and included the following: four patients requiring radiological embolization of an arterio-venous fistula. Right-sided haemothorax following an upper pole renal access to a complex staghorn calculus in a patient who had previous open nephrolithotomy in another European country. This complication was recognized intraoperatively, and a chest drain was inserted, which was removed 48 hrs later. Moreover, one patient sustained a puncture of the descending colon. The patient had a previous retroperitoneal laparoscopic pyeloplasty which led to anatomical distortion.

### 3.4. Factors to Predict Success

As mentioned, 475 (79%) procedures resulted in an effective procedure. Further analysis revealed that stone size (*p* < 0.001), location (*p* < 0.001) and GSS (*p* < 0.001) may predict a successful procedure. In contrast, no association was found between procedure success and patients’ age, BMI or stone density. Moreover, a successful procedure was not associated with complications’ rate, type or severity or the length of hospital stay. Nevertheless, overall, the effective procedures were significantly shorter than those that were defined as ineffective (59.1 ± 38 vs. 73.5 ± 53, *p* < 0.001).

Regarding stone location, Staghorn stones had the lowest procedure success rate and longest operative time. On the other hand, stones situated in the renal pelvis had a significantly higher clearance rate than stones in any other position (*p* < 0.001), (Table 5). Further stratification revealed a significant correlation between stone size, the operation’s length, and the rate of successful procedures. No association was found between stone size, length of stay, or the rate of severe complications. Nonetheless, stratification based on GSS revealed that a higher score is significantly associated with longer operative time, lower effectiveness rate (91.9% vs. 39.3, for GSS1 and 4, respectively) and longer hospital stay (Table 5).

## 4. Discussion

Percutaneous nephrolithotomy (PCNL) is the treatment of choice for large renal calculi [2]. The prone position has traditionally been considered the preferred position to obtain renal access due to increased surface area with a low risk of abdominal visceral injuries. However, in recent years the particular focus has been directed toward the supine position as it confers several potential advantages. These include improved patient positioning, shorter operative time, and decreased risk of respiratory compromise, especially in cardiopulmonary disease or overweight patients. Other reported benefits include decreased radiation exposure and easier ECIRS or simultaneous bilateral endoscopic surgery (SBES) [8,12]. The supine PCNL, especially the Galdakao-modified Valdivia position, allows for a more approachable ureteroscopic access, enabling a combined PCNL and URS for managing complex stone disease [6]. Some of the suggested advantages of the ECIRS approach include better visualization of both the lower and upper urinary tract, which may also allow optimal surgery planning. For example, timely detection of ureteral stones, tight uretero-pelvic junctions, urethral strictures or false urethral passages may affect the choice of surgical strategy and the type of instruments used. Moreover, the simultaneous retrograde endoscopic access may offer timely correction of each step of the renal entry (puncture, guidewire application, dilation of the access tract) [13]; it is also a helpful approach to treating encrusted stents.

In the current study, 27.7% of patients underwent ECIRS. While larger stones were more likely to be treated via ECIRS, stone location was not associated with the combined ureteroscopic approach. Regarding surgical outcomes, the overall success rate was similar between both approaches. However, ECIRS patients were more likely to be stented. One explanation could be that urologists tend to place stents even after uncomplicated URS, often due to wanting to prevent possible complications from postoperative ureteric oedema or to aid the passage of small fragments [14,15,16].

In regard to renal drainage, in the current analysis, a PCN tube was sited after the procedure in 97.3% of patients. The remaining 22 patients (2.7%) underwent a “tubeless PCNL” for stones smaller than 2 cm and GSS of 1–2. Although, according to recent literature, tubeless PCNL is a well-tolerated and effective treatment, the decision of whether or not to place a nephrostomy tube after PCNL still depends on the clinical judgment and complexity of the case and experience of the surgeon. Therefore, this high number reflects the practice and experience of the surgeons working in this centre. On the other hand, a simultaneous double-J stent was sited in 45.3% of cases.

Interestingly, the risk factors for stent insertion among those who underwent ECRIS were similar to those seen in patients who underwent PCNL without ureteroscopic access. Additionally, in the two groups, procedures for renal pelvis stones were more likely to be stented. It seems, therefore, reasonable to assume that the reasons for leaving a stent at the end of a “traditional” PCNL (without a URS approach) are the same as those described for URS (aid passage of small fragments, etc.). Still, there is significant level 1 evidence that stented PCNL decreases postoperative pain, analgesic requirements, urine leakage, and duration of hospital stay after PCNL in a selected group of patients [11,17]. Still, other studies suggest that patients who are stented have inferior quality of life than those who receive a nephrostomy tube or an open-ended ureteral catheter, mainly due to stent-related symptoms [18,19].

In many institutes, the decision whether to stent is based on the discretion of the treating surgeon, probably due to the lack of definitive guidelines regarding the indications. In most cases, the decision is based on unwritten services routines. Moreover, the fact that over a third of the patients were stented, regardless to ureteral access, raises doubts about the real need for stent or its potential benefits. This demonstrates why the use of renal drainage after PCNL should be targeted to comply with evidence-based standards during both “traditional” PCNL and ECRIS. Further study is necessary to examine variation in clinical triggers and overall utilization of the stents.

In regard to surgical outcomes, we found that smaller pelvic stone with low GSS are most likely to be successful, regardless to patients’ BMI, stone density. Yet, none of which were associated with higher rate of complications. These findings provide insight into the predictors of a successful surgery and may aid in the development of strategies for improved patient selection and preparation to surgery. For example, planned staged procedure for large or staghorn stone, or even consideration of other surgical options in case of advanced cases with high GSS. Overall complications rate was 21.7%. The main type of complication was infection in 26.2 of cases. This rate of infections could be explained by the nature of stone treatment as well as the rate of staghorn stone, among other risk factors for infections following PNCL [20]. Prevention of this common complication is very important to decrease the morbidity rate following PCNL. In our institute, we have administrated a routine screen of all patients at pre-operative assessment for infection. Moreover, patients with staghorn calculi are regularly placed on low-dose antibiotics whilst they are on the waiting list for their operation. Lastly, at anaesthetic induction, patients are given Aminoglycoside in the form of Gentamicin 5 mg/kg.

Limitations of this study include its retrospective design. This is mainly due to the absence of data on potential confounding factors and the difficulty of identifying an appropriately exposed cohort and an appropriate comparison group. Moreover, the lack of definitive guidelines regarding indications for stent insertion or ECIRS procedure represents some weakness in our study design. However, these weaknesses are partly overcome by the fact that all operations were performed in a single high-volume tertiary care academic institute. Thus, it is reasonable to assume that clinical decisions were derived from similar clinical judgment and service routines which may also reflect the versatility of this approach.

Lastly, it should be noted that all procedures were performed using a 30-F sheath (“standard PCNL”). Recently, new systems for miniaturized PCNL have been developed to reduce the incidence of common complications while achieving comparable stone-free rates. Nonetheless, the implementation of these techniques requires the acquisition of the appropriate equipment as well as appropriate surgical training. Currently, the choice of tract size is still highly debatable and large; prospective trials are still required to prove the advantage of miniaturized PCNL that will justify the need for special equipment, imaging devices and additional training [21].

## 5. Conclusions

We demonstrate that PCNL in the Galdalko modified Valdivia position in a high-volume centre is safe and efficacious. Patients with smaller stones in the renal pelvis and a low GSS have the highest chance of a successful procedure whereas patients with larger and proximal stones are more likely to be stented. Further study is required regarding the role of ECIRS and the usefulness of tubeless procedures.

## Figures and Tables

**Table 1 jpm-12-01956-t001:** Clinicopathological features for 592 patients who underwent supine PCNL.

Variable			Median (IQR)/N (%)
Age			56 (42–67)
Gender	Male		347 (58.6)
	Female		245 (41.4)
BMI			28 (24.8–32.5)
Stone size (mm)			17 (13–23)
Stone density (HU)			1008 (697–1342)
Side	Lt		321 (54.2)
	Rt		262 (44.3)
Stone Location	Renal pelvis		177 (29.9)
	Single pole		186 (31.4)
		LP	130 (69.9)
		MP	29 (15.6)
		UP	27 (14.5)
	Partial staghorn		33 (5.5)
	Staghorn		82 (13.9)
	Multiple		100 (16.9)
	Encursted stent		14 (2.4)
Guy’s Score	1		172 (29.1)
	2		209 (35.3)
	3		107 (18.1)
	4		84 (14.2)

Abbreviation: BMI—Bodd mass index; HU—Hounsfield unit; Lt—left; Rt—Right; LP—Lower pole; MP—mid pole; UP—upper pole.

**Table 2 jpm-12-01956-t002:** Number of punctures stratified by stone position.

Location	≥2 Punctures (*n* = 27)	ECIRS (*n* = 164)
Upper pole		8 (4.9)
Middle/Interpolar		8 (4.9)
Lower pole	1 (3.7)	36 (22)
Pelvis	2 (7.4)	42 (25.6)
Multiple	10 (37)	44 (26.8)
Partial Staghorn	2 (7.4)	7 (4.3)
Staghorn	12 (44.4)	13 (7.9)
Encrusted stent		6 (3.7)

**Table 3 jpm-12-01956-t003:** Risk factors and outcomes for Endoscopic Combined Intrarenal Surgery (ECIRS).

		ECIRS (*n* = 164)	No ECIRS (*n* = 428)	*p* Value
Age		53.9 ± 16.5	55 ± 15.4	0.43
Gender	Male	100 (61)	247 (57.7)	0.47
	Female	64 (39)	181 (42.3)	
BMI		28.5 ± 5.7	29.2 ± 6.7	0.26
Stone size (mm)		20.3 ± 10.1	17.5 ± 7.3	0.013
Stone density (HU)		992 ± 376	1028 ± 407	0.12
Stone Location	Renal pelvis	42 (25.6)	135 (31.7)	0.16
	Single pole LP MP UP	52 (31.7) 36 (22) 8 (4.9) 8 (4.9)	134 (31.5) 94 (22.1) 19 (4.5) 21 (4.9)	0.156
	Partial staghorn	7 (4.3)	24 (5.6)	
	Staghorn	13 (7.9)	69 (16.2)	0.16
	Multiple	44 (26.8)	56 (13.1)	0.01
	Encursted stent	6 (3.7)	8 (1.9)	0.01
Guy’s Score	1	42 (26.8)	130 (31.3)	0.02
	2	70 (44.6)	139 (33.5)	
	3	31 (19.7)	76 (18.3)	
	4	14 (8.9)	70 (16.9)	
≥1 punctures		6 (4.2)	21 (5.1)	0.65
Effective procedure		125 (76.2)	332 (77.6)	0.72
Stent insertion		105 (64)	153 (35.7)	0.001
Major complication rate (CDC ≥ 3)		7 (4.3)	26 (6.1)	0.59

Abbreviation: BMI—Bodd mass index; HU—Hounsfield unit; LP—Lower pole; MP—mid pole; UP—upper pole; CDC—clavien dindo classification.

**Table 4 jpm-12-01956-t004:** Risk factors for stent insertion.

		Full Cohort (PCNL + ECIRS), *n* = 592			Only PCNL, *n* = 428		
		Stent Inserted (*n* = 258)	No Stent (*n* = 334)	*p* Value	Stent Inserted (*n* = 153)	No Stent (*n* = 275)	*p* Value
Age		55.5 ± 15.7	54.2 ± 15.5	0.33	56.6 ± 15.3	54.2 ± 15.5	0.13
Gender	Male	145 (56.2)	202 (60.5)	0.29	88 (57.5)	159 (57.8)	0.95
	Female	113 (43.8)	132 (39.5)		65 (42.5)	116 (42.2)	
BMI		28.9 ± 6.1	29.2 ± 6.9	0.62	29.1 ± 6.6	29.3 ± 6.8	0.83
Stone size (mm)		21.2 ± 11.5	18.08 ± 7.6	0.001	23.8 ± 13.3	18.5 ± 7.6	0.001
Stone density (HU)		992 ± 24.7	1035.1 ± 25	0.22	989.9 ± 382	1051 ± 420	0.14
Stone Location	Renal pelvis	67 (26)	110 (33.1)	0.12	43 (28.1)	92 (33.7)	0.15
	Single pole LP MP UP	63 (24.4) 44 (17.1) 9 (3.5) 10 (3.9)	123 (37) 86 (25.9) 18 (5.4) 19 (5.7)		32 (20.9) 20 (13.1) 4 (2.6) 8 (5.2)	102 (37.4) 74 (27.1) 15 (5.5) 13 (4.8)	
	Partial staghorn	19 (7.4)	12 (3.6)		13 (8.5)	11 (4)	
	Staghorn	52 (20.2)	30 (9)		42 (27.5)	27 (9.9)	
	Multiple	47 (18.2)	53 (16)		18 (11.8)	38 (13.9)	
	Encursted stent	10 (3.9)	4 (1.2)		5 (3.3)	3 (1.1)	
Guy’s Score	1	60 (24.9)	104 (35.1)	0.001	33 (22.4)	97 (36.2)	0.001
	2	71 (29.5)	125 (42.2)		36 (24.5)	103 (38.4)	
	3	55 (22.8)	42 (14.2)		35 (23.8)	41 (15.3)	
	4	55 (22.8)	25 (8.4)		43 (29.3)	27 (10.1)	
ECIRS		105 (64)	59 (36)	0.001			

Abbreviation: BMI- Bodd mass index; HU- Hounsfield unit; LP- Lower pole; MP- mid pole; UP- upper pole.

**Table 5 jpm-12-01956-t005:** Effectiveness of supine PCNL, stratified by stone position, size and Guy’s score.

**Stone Location**	**Stone Size**	**Operative Time**	**Effective** **Procedure, %**	**Hospital** **Stay (days)**	**Major Complication** **Rate (CDC ≥ 3), %**
Renal pelvis	17.4 + 5.6	55.7 ± 38	92.1	3.4 ± 2.9	4.5
Single pole					
LP	16.8 + 5.9	33.1 ± 3.17	83.1	3.5 ± 2.4	6.9
MP	17.5 + 6.2	77.6 ± 40.1	85.2	3.3 ± 2.3	3.4
UP	18.2 + 6.7	60.55 ± 37.7	69	3.9 ± 2.9	7.4
Partial staghorn		66.3 + 38.9	64.5	2.9 ± 1.5	6.1
Staghorn		86.3 ± 56.7	45.1	5.4 ± 7.9	7.3
Multiple		67.8 ± 43.6	73	3.76 ± 3.8	4
*p* value		*p* = 0.001	*p* < 0.001	0.107	0.360
**Stone Size**		**Operative time**	**Effective** **procedure, %**	**Hospital** **stay (days)**	**Major complication** **rate (CDC ≥ 3), %**
≤2 cm		55.9 ± 35.4	85	3.45 ± 2.7	4.6
2–3 cm		63.75 ± 38.8	78.5	3.44 ± 2.6	8.4
>3 cm		64.5 ± 48.9	55	3.65 ± 2.7	3.2
Staghorn		86.3 ± 56.7	45.1	5.4 ± 7.9	7.3
*p* value		*p* < 0.001	*p* < 0.001	0.45	0.240
**Guy’s score**		**Operative time**	**Effective** **procedure, %**	**Hospital** **stay (days)**	**Major complication** **rate (CDC ≥ 3), %**
1		54.6 ± 30	91.9	3.2 ± 2.2	5.81
2		55.8 ± 37.4	83.7	3.7 ± 3.2	4.31
3		69.1 ± 46	70.1	3.3 ± 2.3	5.6
4		86.3 ± 56.4	39.3	5.3 ± 7.8	8.3
*p* value		0.029	*p* < 0.001	0.039	0.09

Abbreviation: LP—Lower pole; MP—mid pole; UP—upper pole; CDC—clavien dindo classification.

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
