# Peer review of "Factors Predicting Outcomes of Supine Percutaneous Nephrolithotomy: Large Single-Centre Experience"

_jpm, 2022, doi:10.3390/jpm12121956_

Round 1
Reviewer 1 Report
This is a retrospective study of a few hundred patients that have been operated for stone disease with PNL procedure (VG). Despite the fact that the study is well conducted and presented I dont see how this paper can add anything to the existing knowledge. There are tons of studies about this technique with much more patients and there are also big meta analysis concerning this topic. There is nothing novel to report with this relatively small study.
Author Response
Thank you for your comment. But we believe that the current study can add more information and data, specifically about the Galdalko modified Valdivia position. Moreover, it is a large cohort from a single institute with a small number of surgeons, which allows us to avoid the most significant confounders.
Reviewer 2 Report
The authors intended to identify the factors influencing the results of PCNL in supine position, and they did find some related factors. But I have some concerns about this article.
1. In line 50, the authors mentioned "prospective data", it is a retrospective study, how to interpret it?
2. In line 69, patients receiving ECIRS were also included in the analysis. This would influence the SFR of PCNL and comprise the reliability of the results, it would be better to exclude those patients.
3. Encrusted stent has distinct characteristics compared to stone; it is not appropriate to include it in the analysis.
4. In line 115, ultrasonic and pneumatic lithotripter have different influence on SFR, a stratified analysis or multiple analysis would be appropriate to diminish this influence.
5. The mean hospital stay may be influenced by many factors, such as comorbidity. If the authors wanted to evaluate the success of supine position by this variable, it should be adjusted.
6. Varied factors would influence the success and safety of supine PCNL, a significant difference between groups for those factors did not warrant a relationship, regression, or multiple variable analysis should be used to identify related factors.
7. The discussion should focus on the related factors of success and safety of supine PCNL, but not on stent insertion and ECRIS.
8. A comparison with other popular PCNL positions is necessary to highlight the superiority or equivalence of supine PCNL, or else the success and safety of supine PCNL could not be testified. It is not reasonable to reach the conclusion mentioned in line 266 based on the results in this article, it did not correspond to the objective of this paper.
Author Response
Thank you for considering our manuscript for publication in. We are grateful to the reviewers for taking the time to point out several elements that needed clarification and correction; the changes and additions that we made using the reviewer’s comments as guidelines are added in the text.
Point to point response:
- In line 50, the authors mentioned "prospective data", it is a retrospective study, how to interpret it?
Thank you for the comment. Indeed, we meant retrospective. The data was collected prospectively yet the analysis was done retrospectively.
- In line 69, patients receiving ECIRS were also included in the analysis. This would influence the SFR of PCNL and comprise the reliability of the results, it would be better to exclude those patients.
As can be seen in the results, we further analysed the 428 patients who underwent PCNL without a combined URS when indicated.
- Encrusted stent has distinct characteristics compared to stone; it is not appropriate to include it in the analysis.
Only 14 patients underwent surgery due to a forgotten encrusted stent. Inclusion of these patients helped answering questions like indications for ECIRS. Excluding these patients from analysis of the main cohort revealed no significant difference.
- In line 115, ultrasonic and pneumatic lithotripter have different influence on SFR, a stratified analysis or multiple analysis would be appropriate to diminish this influence.
Thank you for this important comment. Ultrasonic and pneumatic lithotripters were the most commonly used energy source in 73% of cases. Unfortunately, we do not have the data available (in regards to which patient was treated in which method) despite revisiting the operating notes. We appreciate that there may be a different but we hope that it is marginal and does not effect the overall message of our study.
- The mean hospital stay may be influenced by many factors, such as comorbidity. If the authors wanted to evaluate the success of supine position by this variable, it should be adjusted.
We appreciate the reviewers comments yet do not believe that further analysis will benefit. The overall mean hospital stay was 3 days and was not found to be associated with any of the other outcomes/variables.
- Varied factors would influence the success and safety of supine PCNL, a significant difference between groups for those factors did not warrant a relationship, regression, or multiple variable analysis should be used to identify related factors.
Thank you for your comment. We agree. A multivariant analysis was performed and revealed similar results.
- The discussion should focus on the related factors of success and safety of supine PCNL, but not on stent insertion and ECRIS.
Given the interesting results in this study w e chose to discuss both.
- A comparison with other popular PCNL positions is necessary to highlight the superiority or equivalence of supine PCNL, or else the success and safety of supine PCNL could not be testified. It is not reasonable to reach the conclusion mentioned in line 266 based on the results in this article, it did not correspond to the objective of this paper.
The authors are not entirely sure to which of the conclusions the reviewer is referring, yet we believe that we managed to demonstrate that PCNL the Galdalko modified Valdivia position, in a high-volume centre is safe and efficacious.
Reviewer 3 Report
Dear authors ,The manuscript is very interesting. Both the study and the analysis
of results are well described. This is a well-designed study and I believe
no major changes are needed. I think that conclusion paragraph must be improved In addition, the authors contribution paragraph should be added.
Author Response
We thank the reviewer for his supportive and kind words. We have added the authors contribution as requested by the journal and amended the conclusions.
Round 2
Reviewer 2 Report
1.According to the responses of the authors, ECIRS didn't pose any influence on the results because of the limited number, so it is reasonable to exclude it, the authors could discuss this topic in another special article.
2.Stent insertion is a result following operation, but not the predictor of outcome, it is not appropriate to include in this article.
3.The results of multivariant analysis should be presented.
Author Response
Thank you for considering our manuscript for publication. We are grateful to the reviewer for taking the time to point out several elements that needed clarification and correction; please find a point-to-point response:
1. According to the responses of the authors, ECIRS didn't pose any influence on the results because of the limited number, so it is reasonable to exclude it, the authors could discuss this topic in another special article.
Simultaneous ECIRS was carried out in 164 (27.7%) cases, which allowed us to examine its role in this setting, hence table 3.
2. Stent insertion is a result following the operation, but not the predictor of outcome, it is not appropriate to include it in this article.
Thank you for your comment, but we included stent insertion with the outcomes (not predictors) along with operative times, surgical effectiveness, and complications (please see methods).
3. The results of the multivariant analysis should be presented.
Thank you for your comment, however, the results were added following the reviewer’s previous comments.